# Remote, Hybrid, and On-Site Work during the SARS-CoV-2 Pandemic and the Consequences for Stress and Work Engagement

**DOI:** 10.3390/ijerph19042400

**Published:** 2022-02-19

**Authors:** Antoni Wontorczyk, Bohdan Rożnowski

**Affiliations:** 1Institute of Applied Psychology, Faculty of Management and Social Communication, Jagiellonian University, 30-348 Kraków, Poland; antoni.wontorczyk@uj.edu.pl; 2Institute of Psychology, Faculty of Social Sciences, John Paul II Catholic University of Lublin, 20-950 Lublin, Poland

**Keywords:** COVID-19, remote work, stress, work engagement

## Abstract

With the COVID-19 pandemic having disrupted economies, businesses, and individual activities, it is important to examine how different forms of work affect employee behaviour. This study applies work engagement (the key construct in organisational psychology) as the dependent variable and considers its determinants in the form of stress factors and attitudes toward remote work. A cross-sectional study was conducted. A total of 544 (Female = 58.5%) workers were surveyed: remote (*n* = 144), hybrid (*n* = 142), and on-site (*n* = 258). The selection for the study was purposive. Standardised survey questionnaires were used in the study: UWES-9, Stress Management Standards, and Attitudes toward Remote Work. The obtained results indicate that there were no significant differences between groups in terms of the intensity of work engagement, but work engagement was explained by other variables that are different in each of the studied groups. Relationships and use of social media were the most important factors among remote workers. For on-site workers, the most important factors were control and role definition. For practitioners, the results indicate which aspects of work should be considered in order to maintain high levels of work engagement when employees are transferring to other forms of work.

## 1. Introduction

Over the last 2 years, the SARS-CoV-2 pandemic has greatly affected many people’s lives. Carnevale and Hatak [1] indicated the complexity of the changes caused by the pandemic concerning not only work itself but also its environment. The International Labour Organisation (ILO) [2] identifies the main additional burdens in this respect, all of which are mainly associated with remote working, to which a large number of employees around the world have been assigned: isolation, blurring of work-home boundaries, and domestic conflicts. However, the analysis of remote work makes it possible to add several other factors to this list, in particular the inadequacy of equipment and organisation of the workplace. Unlike the employer’s workplace, the home does not have adequate equipment and resources. Home equipment is usually not as good as the employer’s, and the places where remote work is performed are mostly adaptations of home conditions and are not properly designed work spaces (ergonomics). Moreover, a remote worker does not have the same access to all the necessary resources (documentation, databases, colleague support, etc.).

### 1.1. Remote Working

The concept of remote work was developed in the early 1970s and simply meant working remotely with the use of IT devices and office equipment. This work was referred to as teleworking. Teleworking is a method of organising and performing work by which the employee works outside the employer’s workplace for a significant part of their working time, providing the employer with work outcomes (results) using information technology and data transfer technologies, especially the internet. In the era of general access to the internet and widespread computerisation, the term “teleworking” has been replaced with the term “remote work”.

Therefore, remote work often happens in a home office [3], which has many benefits. Remote work allows people who would not be able to work in the employer’s workplace to be employed and perform their professional duties. Examples include people raising disabled children, people living far away from the place of employment, experts, and eminent specialists (providing services as part of additional work). Other advantages include (a) the possibility of free contact with relatives, which makes it easier to maintain a balance between work and family [4]; (b) the possibility of reconciling work and personal life (including family life) by spending more time with loved ones; (c) no need to travel from home to work and back every day, thus saving time and money; (d) environmental protection [5].

Studies on remote working have also highlighted positive management effects: (a) greater involvement of employees in their work (e.g., due to better concentration on tasks as a result of the peace and quiet that can be found at home); (b) faster and more efficient performance of duties; (c) greater efficiency and productivity; (d) greater flexibility in planning one’s own activities and adjusting them to one’s daily rhythm and way of working; (e) higher employee loyalty to the employer [6].

A few studies on remote work have been conducted during the SARS-CoV-2 pandemic. Yancy [7] arrived at an interesting result: in his study, remote work was perceived as a privilege that is available to only a few. A study conducted in Wuhan, China, showed that almost 30% of patients who contracted COVID did so in an employer’s workplace [8]. Employees working from home simultaneously have the opportunity to maintain social distancing and self-isolation in the event of infection or exposure to high-risk contacts. This has become the dominant form of work in numerous sectors [9]. In order to contain the spread of the virus, various measures have had to be implemented at work: reducing face-to-face contact, social distancing, adequate ventilation, hand hygiene, use of personal protective equipment, and self-isolation. In this context, remote or hybrid work was one of the most frequently used solutions [10,11,12,13]. Remote work has therefore become not only a privilege but even a necessity to protect employees from infection. Bearing in mind these results, in our study the hypothesis was adopted that perceptions of remote work vary depending on its form (H1). The next hypothesis was that the greatest number of positive effects of remote work are observed in groups of employees who perform their work remotely, slightly less in the case of hybrid work, and least for on-site work (H1a).

On the other hand, several studies have shown that remote work was also perceived as negative by employees [2]. This applies in particular to people who contracted COVID-19 but still had to work remotely. Employees performed their duties because they were concerned about job insecurity and the lack of social protection [14,15]. Therefore, a hypothesis was adopted that employees performing remote work its negative effects, in contrast to employees performing work in the traditional form (H1b), in contrast to employees performing work in the traditional form [14,16].

As shown in many studies on the SARS-CoV-2 pandemic, employees who could not perform their professional tasks remotely found themselves in a difficult situation not only due to greater exposure to infection but also due to (a) pressure from supervisors to continue working, even with flu-like symptoms [17]; (b) little possibility of obtaining paid sick leave [18]; (c) performing work in conditions that do not comply with health and safety regulations in the context of the pandemic [15,19].

The SARS-CoV-2 pandemic situation is particularly difficult for workers with poor economic status and lack of job security [20], i.e., expatriates or temporary workers. Failure to work under the conditions imposed by an employer has sometimes resulted in deportation or loss of residence permits [21,22]. This situation was experienced especially by women, not only first-line staff working with the sick, but also those in other sectors of the economy. As research results show, women were paid worse than men, discriminated against at work on the basis of gender, and even abused [21,23,24,25,26]. In addition, on-site work during the pandemic has led to an increase in the perception of somatic ailments, such as fatigue, stomach upsets, sleep disturbances, headaches, or chronic pain [27].

In the situation of remote work, respondents indicated the need to quickly adapt to the new reality of the work organisation and bring work and family tasks into harmony [10,11,28]. In the situation of remotely performed work, there was also a change in the dynamics of teamwork, management methods, and social communication, all of which directly influenced the effectiveness of employees [29]. In the literature, attention is also paid to the discrepancy between the expectations and the real challenges of remote work. The dimension of the employee’s control has changed, therefore it can be assumed that employees must gradually get used to working remotely: they should perceive it not only as a necessity in an emergency, but also as a natural change as a result of the Information and Communication Technologies (ICT) transformation. Therefore, the hypothesis was adopted that employees working remotely perceive their form of work as an inevitable necessity that not only protects them from infection but also facilitates and supports the work process (H1c).

The pandemic situation and remote work have significantly increased the feeling of job insecurity [30]. Moreover, changes related to the necessity to work remotely have been imposed, regardless of employees’ preferences. This has negative consequences for employees, such as difficulties in disconnecting from job requirements, separating work from private life, and even other psychosocial risks such as isolation [2,31]. In view of the broad context of the possible consequences of remote or on-site work during a pandemic, it is important to investigate how the pandemic situation and the related forms of work translated into employees’ functioning and perceived stress. Therefore, crucial aspects of employees’ behaviour need to be described as they allow work performance to be perceived from various angles: performance level, satisfaction, job crafting, etc. This concept also includes “work engagement” [32,33].

### 1.2. Work Engagement

In the last 20 years, the term “work engagement” has become very popular. The EBSCO database (accessed on 28 March 2021) lists 40,645 publications in scientific journals in which “engagement” is a Keywords: In the last 5 years, the number of publications was almost 2.5 times higher than for the previous period (2017–2021: 28,937: 2012–2016: 11,708). This popularity is due to the fact that work engagement seems to be significantly positively related to adaptation to changes [34], job crafting [35], and many constructs that describe employee behaviour, such as work efficiency, creativity, civic behaviour in the organisation, and customer satisfaction [36,37]. Numerous studies have attempted to define work engagement. W. Kahn [38] used the term “personal engagement”, which he defined as an employee’s commitment to work in the physical, cognitive, and emotional spheres. The physical sphere is expressed in the work of the employee; the cognitive sphere is expressed in the level of concentration; the emotional sphere is expressed in emotional activation during the performance of tasks. Kahn pointed out that a prerequisite of employee involvement is the sense of job security, which is also important if employees perceive their job as meaningful and are autonomous in their work. R. Kanungo [39] defines commitment to work as the cognitive and cognitive-emotional state of mental identification with work. He also distinguished the English terms “Job Involvement” and “Work Involvement”. According to him, the former term refers to involvement in a specific job that a person is currently doing; the latter refers to involvement in work understood as a sphere of human activity, regardless of its specificity. Involvement in a specific job is related to the degree to which it meets a person’s current needs. The second definition of commitment refers to treating work as valuable and is associated with the construct of work centrality. This type of commitment is the result of the socialisation process, which emphasises the importance of work.

The most popular approach in scientific research focused on work engagement is that of Schaufelli and Bakker [40], who view work engagement as a positive affective, motivational, rewarding, and work-related state of mind. Baker and Demerouti [32] called it Job Demands–Resources Theory (JD-R). According to them, the symptoms of commitment are a high level of energy combined with a high level of dedication and a strong focus on work. The source of this state is the balance between requirements and labour resources. As Bakker and Demerouti [32] claim, the most significant predictors of work engagement are work resources (e.g., support, autonomy, feedback) and personal resources (e.g., positive self-evaluations, self-efficacy). In line with Hobfol’s theory [41], resources are understood as something that is valued in itself or which acts as a means to achieve important goals. Work resources are the physical, organisational, and social aspects of a work situation that help to achieve work goals. On the one hand, resources minimise the negative impact of requirements; on the other hand, they favour the individual development of employees [42]. As a part of the work resources-requirements model, the following personal resources were distinguished: self-efficacy [43], self-esteem based on organisation [44], and optimism [45]. However, these can be broadly defined as a general dimension that refers to individuals’ perceptions of their ability to meet requirements in a wide range of contexts [46].

As indicated at the beginning of the article, in remote work, compared to work in the workplace, there are many factors that make it difficult to perform work; these change the resource-requirements relationship and shape completely different conditions for employee involvement. Research indicates a very wide range of responses to increased levels of psychosocial stress [47]. It is necessary to examine whether the changes brought about by even a partial transition to remote work affect the level of engagement in work. Therefore, the exact requirements of on-site, hybrid and remote work need to be analysed. Recent studies suggest that despite the convenience and availability of electronic devices [48,49], remote work is considered inferior to on-site work [48,49]. Additionally, in the population of Polish employees, remote work before the pandemic was rather uncommon, except for in some industries, e.g., IT. Therefore, due to its novelty and the weaknesses described in the aforementioned research, we hypothesised that on-site workers’ engagement is higher than that of those working remotely or in a hybrid manner (H2).

### 1.3. Stress at Work

The drawbacks of remote work described in the literature are generally due to the greater stress generated in this situation [48,49]. Therefore, we hypothesised that the stress of remote workers would be higher than that of on-site workers (H3). It is also worth investigating stress in terms of more basic elements that are analysed separately. Job requirements in the JD-R theory are described as factors that modify (moderate) the relationship between resources and commitment. Bakker and Demerouti [32] enumerate the following work requirement factors as examples: high work pressure, an unfavourable physical environment, and emotionally demanding interactions. In earlier studies underlying the JD-R theory, R. Karasek [50] identified the main work stressors as the main examples of work requirements: work overload, time pressure, role conflict and control of the work situation. Similarly, in their Work-Life Areas model, Leiter and Maslach [51] mention six factors important for burnout and work engagement: Workload, Control, Reward, Community, Fairness, and Values. However, subsequent studies showed that the impact of Work-Life Areas on burnout and commitment turned out to be very complex [52]. In all of the models cited, the key concept—at a higher level of generality—that characterises job requirements is “stress”. This is a key concept in occupational psychology, which is interested in distress or eustress. The former focuses on negative work-related phenomena (e.g., burnout); the latter focuses on positive phenomena (e.g., engagement, civic behaviour).

The Health and Safety Executive (HSE) made an interesting attempt to describe work-related stress [53] that assumes the key stressors at work to be (1) “Demands” (workload, work patterns, and the working environment); (2) “Control” (how much say the person has in the way they do their work); (3) “Support”—Managers’ support + peer support at work (which includes encouragement and resources provided by the organisation, line management, and colleagues); (4) “Relationship” (which includes promoting positive working practices to avoid conflict and deal with unacceptable behaviour; (5) “Role” (whether people understand their role within the organisation and whether the organisation ensures that the person does not have conflicting roles); (6) “Change” (how organisational change is managed and communicated). The stressors that are most compatible with the models of Karasek are “Demands” (1), “Control” (2), “Role” (5), and “Change” (6). These stressors cover almost the entire spectrum of possible sources of high work pressure, an unfavourable physical environment, and emotionally demanding interactions, as described by Bakker and Demerouti [32]. On the basis of the JD-R theory, “Support” (3) and “Relationships” (4) should instead be classified as labour resources. The source of negative stress in this case is low levels of these resources.

Demands include factors such as workload, work patterns and the working environment, all of which translate into worse perceived stress [32]. With regard to remote work, it can be assumed that because it causes an overlapping of professional and family obligations, the employee feels more overloaded (Gabr et al. 2021). At the same time, the change of workplace and difficulties related to contact with co-workers create a feeling that the previously used work strategies are inadequate and force the need to experiment with new strategies, which also makes it difficult to perform work. Additionally, since the home environment is not designed for professional work, it causes many inconveniences: especially the need for quick contact with colleagues, but also working conditions, the speed of internet connections, and the condition of equipment, all of which cause stress [54]. The home environment includes many psychosocial distractors, such as the presence of competing roles (e.g., parent) that generate additional tasks and responsibilities for the employee. As indicated in the ILO report [2], work difficulties resulting from remote work can generate anxiety and depression.

Therefore, we proposed a hypothesis (H3a) that the level of work demands is higher in the remote worker group than in the on-site worker group. The level among employees working in a hybrid system will be similar to that of remotely working employees. Moreover, the JD-R model shows that work demands have a significant negative relationship with work engagement.

One of the important characteristics of work situations that influences organisational behaviour is the interpersonal relationships that prevail within the company. These include feelings of fair treatment, trust, and kindness [55], but also rivalry, interpersonal conflict, and related company policies [56]. Research indicates that a pandemic situation affects interpersonal relationships within an organisation [57], and remote working modifies the shape of relationships between workers [58]. Therefore, we hypothesise that remote workers experience less of the negative aspects associated with interpersonal relations (H3b).

According to Karasek [50], in addition to work demands, “control” is one of the two main factors influencing employees’ response to work. The literature review shows that control can be understood in three ways: a feature of a work situation, a personality trait of an individual, or a subject’s conviction about their influence on the work environment. The latter understanding is most interesting in the context of work. It is understood as an employee’s belief in how much say they have in the way they do their work. Research shows that the sense of control in the workplace mainly impacts occupational stress and occupational burnout [59], but it also has an impact on employees’ feeling of job satisfaction and selected dimensions of mental health. Remote work involving the physical (geographical) detachment of an employee from the workplace causes a loosening of organisational ties. The supervisor can no longer enter the room where the employee works whenever he or she wants to: they can only contact the employee remotely. However, remote work may also encourage superiors to inspect employees at any time of day. Therefore, the employee is obliged to plan and control the results of his work, which translates into the belief that the work situation is better. In P. Warr’s Vitamin Model [60], control is an AD vitamin: both a shortage and an excess of it are a source of dissatisfaction with work. However, since it is the remote worker who assesses himself, it was assumed (H3c) that remote workers perceive the level of control of work as more suited to their needs.

According to Karasek [50], support is another factor that models the amount of work stress. In the perception of work stress, Karasek emphasised the role of social interactions and thus the form of direct relationships with others. Research by [61] suggests that the relationship with the supervisor plays the dominant role. Superiors’ support influences work attitudes and organisational commitment [62]. However, Treiber and Davis [63] indicate that it is also important to gain support from other co-workers, which can also be a source of companionship and is particularly important during teamwork [64]. In both these cases of remote work, due to the remoteness of the resulting form of work, the form of real relationships with colleagues (either managers or colleagues) is changing (which includes dealing with unacceptable behaviour and promoting positive working practices to avoid conflict). However, support can also be understood as a systemic solution in organisations as it is an element of organisational culture that consists in the encouragement and resources provided by the organisation, line management and colleagues [53]. It is based on the empowerment of employees. It can be expected that due to the distance from the organisation, which reduces direct contact, support from management and colleagues in the situation of remote work is weaker in comparison to on-site employees (H3d).

Another stress factor often indicated in the literature is job role, which is defined as the extent to which employees know what to do and what not to do. This phenomenon may consist in a lack of information about the content of the employee’s role, or contradictory information from many sources, or from the same source but in different situations [38]. A special example of stress relative to job role is internal role conflict, which consists in the need to act against one’s own values in order to achieve job-related goals (whether people understand their role within the organisation and whether the organisation ensures that the person does not have conflicting roles). As the scope of superiors’ control in remote work is smaller, a remote employee’s freedom to define their own role is greater, thus stress related to this aspect is lower (H3e).

Change (how organisational change is managed and communicated). Many models of employee functioning emphasise the importance of autonomy at work [65], which is sometimes called control of the work situation [50]. Therefore, it can be assumed that an employee may be transferred to remote work mode by his superiors without asking for his opinion. In this situation, this remote worker treats the change as something negative, which will be an obstacle to them feeling engaged in work (H3f).

It is interesting to look at the model of shaping work engagement in respect to on-site, hybrid, and remote work, and to compare these results between groups. Due to the fact that the model of the determinants of engagement is very complex, it can be assumed that completely different aspects of work will be important in the case of employees working in remote or on-site mode. A particular work demand that is relevant to one form of work may be irrelevant to another, in part because of differences in the intensity of work demands, which have been the subject of earlier hypotheses, and because of the different situational context. Therefore, we hypothesised that the micromodel of the determinants of work engagement differs according to the form of work (H4). In our research, there will be three common forms of work: on-site, hybrid (combined on-site and remote work), and remote.

## 2. Materials and Methods

### 2.1. Problem and Hypothesis

The presentation of the research method requires a clear formulation of the research focus and problem. The research question of our project is whether the groups (remote, hybrid, and one-site work) differ in terms of work engagement, stress, and the links between these variables. For this purpose, all the hypotheses posed and justified in the theoretical introduction are collected at the beginning of this section. This makes it easier to understand what is being researched and to see that the research method is well suited to the problem. The subject of our empirical analysis is to check the validity of the following four main hypotheses and some of their details:

**H1.** 
*Perceptions of remote work vary depending on its form.*


**H1a.** 
*The greatest number of positive effects of remote work are observed in groups of employees who perform their work remotely, slightly less in the case of hybrid work, and least in the on-site form.*


**H1b.** 
*Employees performing remote work also see its negative effects, in contrast to employees performing work in the traditional form.*


**H1c.** 
*Employees working remotely perceive their form of work as an inevitable necessity that not only protects them from infection but also facilitates and supports the work process.*


**H2.** 
*On-site workers’ engagement is higher than those working remotely or in a hybrid manner.*


**H3.** 
*The stress of remote workers is higher than that of on-site workers.*


**H3a.** 
*The level of work demands is higher in the remote worker group than in the on-site worker group.*


**H3b.** 
*Remote workers experience less of the negative aspects associated with interpersonal relations.*


**H3c.** 
*Remote workers perceive the level of control as more suited to their needs.*


**H3d.** 
*Due to the distance from the organisation, which reduces direct contact, support from management and colleagues in the situation of remote work is weaker in comparison to on-site employees.*


**H3e.** 
*Remote employee’s freedom to define their own role is greater, and stress related to this aspect is lower.*


**H3f.** 
*Remote workers treat the change of work form as something negative, which is an obstacle to them feeling engaged in their work.*


**H4.** 
*The micromodel of the determinants of work engagement differs according to the form of work.*


### 2.2. Questionnaires

The UWES questionnaire, which is currently the most widely applied in the world, was used to study the key dependent variable, namely work engagement [40]. It has two versions: a 17-item version and a shortened nine-item version. Validation studies show that the short version has better psychometric indicators, therefore this version was used in our study. It has a three-factor structure: Vigour, Dedication, and Absorption. Each factor is measured by three theorems (Cronbach’s Alfa). The respondents use a seven-point scale to state how often the phenomenon described by each statement occurs in their work.

The measurement of variables related to the characteristics of the work situation was conducted using the “Management Standards” questionnaire [66]. This questionnaire consists of 35 statements that measure seven key work-related stress factors: Demands—issues such as workload, work patterns, and the work environment; Support—the encouragement, sponsorship, and resources provided by the organisation, line management and colleagues; Relationships—promoting positive interpersonal relations to avoid conflict and deal with unacceptable behaviour; Control—how much say the employee has in the way they do their work; Role—whether people understand their role within the organisation and whether the organisation ensures that the person does not have conflicting roles; Change—how organisational change (large or small) is managed and communicated in the organisation. The reliability of the individual scales of the questionnaire has been confirmed (0.66 < α Cronbach’s < 0.84).

Attitude towards remote work was measured with the Remote Work Test (Test Pracy Zdalnej, TPZ) questionnaire by Bartczak and Wontorczyk [67]. This questionnaire consists of 35 items measured on a five-point scale, where 1 means “I strongly disagree” and 5 means “I strongly agree”. It includes three factors related to remote work: positive reinforcement, negative reinforcement, and temporal orientation. This tool has good measures of internal reliability, as measured with Cronbach’s alpha coefficient: ranging from 0.82 for Temporal Orientation to 0.92 for Positive Reinforcement. A shortened 11-item version of this questionnaire was used in this study. Four items were for positive reinforcement; four were for temporal orientation; and three were for negative reinforcement. The internal reliability of the subscales of the shortened version of TPZ turned out to be slightly weaker than the full version, but it was satisfactory. Cronbach’s alpha coefficient for the subscale was 0.82 for Positive Reinforcement, 0.72 for Negative Reinforcement, and 0.70 for Temporal Orientation. The Positive reinforcement scale describes the positive aspects of remote work, indicating such issues as time saving, greater availability for the family, better organisation of working time, and greater possibility of carrying out tasks outside of work. In turn, negative reinforcement scale indicates the negative dimension of remote work, lack of contact with superiors, a sense of uncertainty regarding the quality of work, as well as further employment, lack of contact with colleagues and management, and conflicts in the family. Temporal orientation describes the perception of remote work as a necessity to which one should adapt, due to not only the circumstances of the SARS-CoV-2 virus but also new challenges related to the organisation of work in the contemporary digital reality.

In addition, in order to also measure physical work conditions, individual questions were asked in the demographics part of the research questionnaire about housing conditions and internet connection speed. To measure time pressure, there were also questions about checking official e-mail accounts and contacting superiors outside working hours.

### 2.3. Sample

The research was carried out online at the beginning of April 2021, i.e., at the peak of the third wave of SARS-CoV-2 in Poland, when the number of COVID-19 cases exceeded over 30,000 people a day. The research sample consisted of a total of 533 respondents living in three provinces in southern Poland: Małopolskie, Świętokrzyskie and Podkarpackie. One hundred and thirty-nine people worked remotely, 140 were hybrid workers, and the remaining 254 people worked from their employers’ offices. The study group included 312 women (58.5%) and 221 men (41.5%). Of the respondents, 42.4% had secondary or primary education, and the remaining 57.6% were people with higher education. 96 people (18%) were employed in managerial positions, 244 (45.8%) worked as specialists, and 193 respondents performed simple work (executive positions). The sample was deliberately selected to ensure large numbers of respondents in both forms of work: remote (*n* = 286) and on-site (*n* = 258). Then, the group of remote employees was divided into two subgroups: only working remotely (*n* = 144); hybrid employees, i.e., those combining on-site and remote work (*n* = 142). These three groups became the basis of our analysis.

The study was conducted in accordance with the Declaration of Helsinki; it was approved by the Institutional Ethics Committee of the Institute of Applied Psychology of Jagiellonian University (protocol code 109/2021, 30 November 2021) for studies involving humans. Data available in the Institute of Applied Psychology Jagiellonian University, Department of Work and Organizational Psychology

## 3. Results

This section is divided by subheadings as this should provide a concise and precise description of the experimental results, their interpretation, as well as the experimental conclusions that can be drawn.

### 3.1. Descriptives Demographic

The demographics data in Table 1 shows that in 33 cases (6.2%) the participants’ housing conditions “did not allow them to work from home”, and 54 respondents (10.1%) “were probably not allowed to work from home”; however, most of the respondents said they “were definitely allowed to work from home” (*n* = 142; 26.6%) or were “probably allowed” (*n* = 181; 34%). The rest of the sample responded without making a clear declaration (*n* = 123; 23.1%). At the same time, most of the respondents declared that they had good internet bandwidth (*n* = 376; 70.5%).

When describing their internet activity, 476 (90%) respondents stated that they had an account on one or more social networks. A lack of such a profile was declared by the remaining 54 participants (10%). One hundred and sixty-four respondents (30.8%) answered work e-mails containing tasks to be performed after working hours. Ninety-nine respondents reported that they received e-mails from supervisors’ containing only work-related content (18.6%). The other 270 respondents did not answer work e-mails at home after working hours (50.7%), which indicates that some remote workers had established a clear boundary between work and home. At the same time, when asked about contact from their superiors after working hours, some respondents (*n* = 249; 46.7%) replied that they did not respond in matters not related to work (*n* = 56; 10.5%). Two hundred and twenty-eight respondents (42.8%) stated that they were sometimes contacted by their superiors after working hours, but only for urgent matters.

### 3.2. Comparison of Means

The first group of hypotheses was related to the differences between remote, hybrid, and on-site employees in terms of their involvement in work and assessment of working conditions. In order to verify these hypotheses, a series of one-way analyses of variance was performed. The results are shown in Table 2, Table 3 and Table 4.

As can be seen from the data presented in Table 2, there was no significant difference between the groups of remote, hybrid and on-site employees. It can be said that in terms of both the general engagement index and individual factors, these groups do not differ from each other.

However, the analysis of mean work demands showed statistically significant differences between groups only for two variables (see Table 3). These differences concerned the employee having control over the work situation, i.e., the Control variable (F = 4.62; df = 2/530; *p* = 0.01). The lowest results in this respect were obtained by on-site workers (M = 21.26; SD = 5.09). Post hoc analysis with Dunnett’s T3 test (which does not assume homogeneity of variance) indicated that a significant difference was found for hybrid and on-site employees (*p* = 0.01). The difference between the group of remote and on-site employees did not reach the level of statistical significance, although it approached it. The second significantly differentiating variable was the assessment of the Relationships variable (F = 3.11; df = 2/530; *p* = 0.05). The lowest results in this respect were obtained by hybrid employees (M = 7.72; SD = 2.60). This group differs significantly from on-site workers (*p* = 0.02).

The most diverse assessments between the studied groups occurred in terms of responses to the TPZ questionnaire. These data are presented in Table 4. In this case, all the dimensions of the attitude described in the test tool have been rated differently. Remote workers perceive the highest number of positive aspects (Positive reinforcement) of this kind of work (M = 12.61; SD = 2.96). In the remaining two groups, the results obtained from the employees are lower (F = 7.99; df = 2/530; *p* = 0.001). The differences are significant for the groups of remote and on-site employees (*p* = 0.004) as well as remote and hybrid employees (*p* = 0.001). In addition, in terms of perceiving negative aspects, the groups differ significantly (F = 24.13; df = 2/530; *p* = 0.001). In this case, remote workers also perceive the highest number of negative aspects (Negative reinforcement) of their form of work (M = 9.35; SD = 2.15). However, as is easy to see, this is a much lower result than the positive aspects. Compared to hybrid and on-site mode, this difference is very significant: in both cases, it reaches the significance level of *p* < 0.001. The difference between hybrid and on-site employees is also significant (*p* = 0.01). This creates a coherent picture in which the less remote work there is, the less its negative aspects are noticed. In addition, in TPZ Temporal Orientation, the mean obtained by employees in this group is the highest (M = 16.62; SD = 2.39). These results differ from those obtained in the other groups (F = 24.83; df = 2/530; *p* = 0.001). In this case, the groups of remote and hybrid workers are no different. The group of on-site workers gained significantly lower results (M = 11.08; SD = 2.51) than the other two (*p* < 0.001 in both cases). Based on these analyses, it can be said that remote workers see many positives of their work mode; although on-site workers have much less to say about remote work, their cognitive representation of remote work is poorer in all aspects.

### 3.3. Regression Analysis

Quantitative comparison of regression models is very difficult because it requires the comparison of all relationships in the model, therefore the concept of qualitative comparison of models performed separately for individual groups was adopted. A block regression analysis with the input method (SPSS 21) was used. The first block included demographic variables (gender, age, job position, and seniority in the current position); the second block included features related to the use of the internet at work; the third group included characteristics of the work situation, measured with the HSE questionnaire; the fourth group included attitude towards remote work.

The regression analysis results for the group of remote workers are presented in Table 5. Data on demographics and internet use explain a very small part of the variance of the work engagement variable (8%). Only the inclusion in the model of variables describing the work situation increased the percentage of the explained variance to 30%. The addition of data from the TPZ Questionnaire increased this percentage to 32%, which is on the borderline of significance.

In the first block of data, the position (β = 0.180; *p* = 0.035; r_a(b,c)_ = 0.174) and seniority in the current position (β = 0.372; *p* = 0.004; r_a(b,c)_ = −0.243) were linked with work engagement. This shows that higher position and work seniority are linked with a higher level of work engagement. Gender and age showed no relationship with the dependent variable.

The block of variables describing internet use turned out to be significant only in combination with the characteristics of the work situation. In Model 3, the following factors were important: reading emails (β = −0.179; *p* = 0.032); having an account on social networks (β = 0.249; *p* = 0.006); the aspect of work Control (β = 0.253; *p* = 0.006); and Relationships (β = −0.323; *p* = 0.001). Reading emails after working hours was associated with lower work engagement, while having social media account/s was associated with a higher level of it. In the case of control, the higher the sense of influence on the work situation, the greater the commitment. Bad relationships in terms of conflicts in the organisation and friction between employees were associated with lower involvement. In the block of variables related to attitude towards remote work, the “Temporal orientation” aspect of remote work and level of work engagement (β = 0.161; *p* = 0.05) are significant. When employees are able to independently organise their work, this promotes work engagement.

In the group of hybrid employees, demographic variables explained only 4% of the variance; adding another block increased this parameter to 7%; this change was close to statistical significance. However, only adding the third block increased the explained variance to 15%. The addition of the fourth block did not change this parameter, which indicates the low importance of this block in the group of employees combining remote and on-site work. In this analysis (see Table 6.), the demographic variable associated with engagement is gender (β = −0.20; *p* = 0.020). Women were more engaged. There was also a weak relationship with the position held (β = −0.156; *p* = 0.076): the higher the position, the stronger the tendency to reveal commitment to work. After adding the use of the internet block, an important relationship was reading work emails after working hours (β = −0.186; *p* = 0.051). The block of variables of stress factors showed “Role” (role-related work characteristics) (β = 0.213; *p* = 0.037) and “Control” (β = 0.206; *p* = 0.049) as being linked to work engagement. “Role” (including items such as better understanding of one’s tasks and lack of conflicts related to the role) contributes to work engagement. In the case of control, it was the same as in the group described earlier: if employees perceived that they had an influence on the work situation, they were more engaged.

For the group of on-site employees (see Table 7.), the level of explained variance was similar to that of the group of remote employees: 33%. The demographic block allowed only 4% to be explained. Adding a block of variables related to the description of the work situation greatly increased this parameter to 33%. Adding the fourth block of variables, which relates to attitudes towards remote work, did not increase the level of explained variance. Among the demographic variables, gender (β = −0.121; *p* = 0.053; r_a(b,c)_ = −0.120) and position (β = 0.174 *p* = 0.007; r_a(b,c)_ = 0.168) were linked with work engagement. The data showed that women and senior employees were more engaged in their work. After adding a block of variables describing the work situation, a significant relationship occurred between involvement and Role (β = 0.268; *p* = 0.001; r_a(b,c)_ = 0.203), Control (β = 0.331; *p* = 0.001; r_a(b,c)_ = 0.269), and Management support (β = 0.180; *p* = 0.019; r_a(b,c)_ = −0.121). Better understanding of one’s professional role, the ability to influence one’s work, and having supportive supervisors were associated with on-site employees’ greater involvement with their work.

## 4. Discussion

When starting this study on work engagement among employees during the SARS COV-2 pandemic, it was assumed that it may be related to the resulting diversification of work modes. Studies claim that high work engagement is associated with continuous positive motivational feelings and even enthusiasm for one’s job [68]. Employees feel so energised and proud of their work that they often do not control their working time [69]. However, when there are additional non-job-related tasks in the course of work, such as when carrying out professional tasks at home [70], a conflict between work obligations and family obligations may occur. It can be assumed that commitment to work consequently decreases. Hypothesis H2 was thus not confirmed. No statistically significant differences were found in terms of both general and specific forms of work engagement. Employees were equally involved in the performance of professional duties, regardless of whether their work mode was on-site, hybrid, or remote. Interpretation of this relationship should be sought in two directions. The first is related to the good organisation of professional duties performed at home. Workplace management plays a special role here as it allows employees to flexibly manage their work environment [71]. Many studies have found that flexible work arrangements (FWA) improve the well-being and health of employees [72] and even increase work engagement [62,73,74]. A certain exception is the study by Rudolph and Baltese [75], which found that flexible working conditions and organisation had a positive effect on employee work engagement, provided that they were not associated with deterioration of health. Isolation and avoiding contact with management and co-workers through hybrid or remote work prevented employees from becoming infected and thus strengthened the maintenance of health. The obtained results of our study are therefore consistent with these studies. To some extent, these results are also in line with other studies which have shown that work engagement has a positive impact on family life, while workaholism leads to family conflicts and thus has a negative impact [36,76]. Working remotely is conducive to maintaining positive family relationships.

The second way of interpreting the results may be related to the strengthening role of social media, which enables interactivity and openness to social relations in real-time mode. Several studies have confirmed that social media can replace the real social processes that occur in organisations: communication, relationship building, cooperation [77,78,79,80], and work engagement [81]. Other studies have shown that active presence on social media also reduces stress at work and even occupational burnout. In our study, as many as 90% of employees declared that they had social media accounts, which could also contribute to the strengthening of social relations with co-workers or management in remote or hybrid work situations.

Our study also attempted to find connections between the form of work environment and stressors at work (H3). Several hypotheses were formulated in relation to this issue. It was assumed that remote and hybrid work would limit demands (H3a). The results of the study did not confirm these assumptions, therefore the following hypotheses were not confirmed: H3d, which assumed weak support from management and colleagues in remote working situations; hypothesis H3e, which concerns the Role variable; and hypothesis H3f, which concerns the Change variable. Statistically significant relationships were found only in the case of two stressors: Control (H3c) and Relationship (H3b). Regarding control, it was assumed that remote or even hybrid work would be more conducive to the feeling of work control than on-site work. In the case of remote work, employees have a greater ability to control their work, both when its organisation is imposed on them and when they are left to make their own decisions. These results are consistent with the assumptions of the theories of Karasek [50], Warr and Clapperton [60], Bakker and Demerouti [32], as well as with the results of other studies [59]. The interpretation is more complex in the case of the second important regression path, which indicates a positive relationship between employees and management in the case of on-site and remote work, and weaker ones among hybrid workers. In the case of on-site work, frequent and positive encounters with supervisors seem logical, provided that employees experience favourable treatment from management, as pointed out by other researchers [82]. The issue concerning the importance of communication between employees and management is also raised by the LMX theory, the validity of which has been confirmed by the results of numerous studies. Several studies have found that employees with high-quality LMX relationships experience higher autonomy at work [83], are more responsible employees [84], and are more likely to speak out about work organisation [85]. In the case of remote work, it can be presumed that factors that contribute to the perception of good relations with superiors despite the lack of physical contact will be the perception of the employee’s obligation to undertake constructive change [1,86,87]; increased sense of responsibility to perform tasks in difficult new conditions [88]; and behaving responsibly at work [89]. In turn, hybrid work organisation forces employees to constantly adapt to the changing work environment, which contributes to the deterioration of relations with superiors and colleagues. This form of work, as confirmed by observations, is conducive to the emergence of conflicts between employees over the question of who will work in the employer’s office and who will work remotely. The resulting feeling of inequality, both in terms of distribution and information, is an important stressor at work [90,91]. As shown by the results of some studies, perceiving a lack of distributional justice and a lack of informational justice may consequently lead to a decline in work engagement. Admittedly, our research did not find differences in work engagement due to the three different forms of work, possibly because hybrid and remote workers treated these solutions as temporary during the lockdown period.

Finally, an important aim of the study was to find a relationship between perceptions of remote work depending on whether the respondents’ work is performed in the office, in a hybrid manner, or remotely. Statistically significant results were obtained in all three scales of perception of remote work (H1). As expected (H1a, H1b, and H1c), the highest results in all three scales were obtained for people working remotely. Employees perceive all three dimensions of remote work (Positive, Negative, and Temporal). These results are logical because employees who work remotely see its benefits to a greater extent than hybrid workers: protection against infection, flexible organisation of working time, time saving, lower costs, and greater opportunities to pursue hobbies. They understand its limitations (lack of physical contact with superiors and co-workers, job insecurity). Finally, they realise that developing specific organisational strategies for remote work will inevitably make it a reality after the end of the SARS-CoV-2 pandemic. A similar attitude was also detected in employees working in hybrid mode, except for poor perception of the positive aspects of remote work measured by the TPZ questionnaire. In all three aspects, remote work was rated worst by on-site employees. Thus, the obtained results are consistent with other studies that relate to both the positive and negative aspects of remote work. Many studies have found that remote work is a privilege that protects an employee against contracting COVID-19 [7,9,10,11,13]). Its negative consequences were also noticed: working despite being ill [2]; fear of losing a job and income [14,16]; and little possibility of getting paid sick leave [18]. A question arises as to why the employees in our study who continued to perform their professional duties from the office during the SARS-CoV-2 pandemic did not perceive the value of remote work. After all, many studies have indicated the negative effects of on-site working in a pandemic situation: depression, stress, and fear of getting sick [4,92,93,94], and a decline in psychological well-being [95,96]. The above-mentioned research was carried out mainly among broadly understood medical staff: nurses, paramedics, orderlies, and doctors. These people were not only susceptible to infection but also came into direct contact with people infected with the SARS-CoV-2 virus. They were not able to perform their professional duties in any other form. Our research did not involve people working in hospitals, therefore our participants did not experience the stress of working with COVID-19 patients, as has been observed in nurses and doctors [97,98,99,100]. It is worth emphasising that respondents more often indicated all three aspects of remote work measured by the TPZ questionnaire as important when they longer performed it. This regularity was not observed in on-site employees and was observed only weakly in hybrid employees.

Another important aim of the study was to detect predictors of particular ways of performing professional duties among the various independent variables included in the study (H4). As already mentioned, the explanatory variables (potentially assumed predictors) were divided into four groups: (a) demographic variables; (b) having an internet connection at home, its speed and bandwidth; (c) characteristics of the work situation measured with the HSE questionnaire; and (d) attitude towards remote work. We will start the analysis of predictors with the case of employees who perform their work only on-site, i.e., the most unfavourable situation from the point of view of the potential risk of contracting the SARS-CoV-2 virus. The largest increase in the explanatory variance for this form of work was obtained when taking into account the total of the first three groups of variables (a, b, and c). When the fourth group of variables (d) was included in the analysis, the value of the variance of explaining the dependent variable decreased. This result is logical because employees working on-site had virtually no experience of working remotely. The predictors turned out to be control, role, and management support. These results are consistent with other studies which indicated that motivation and job satisfaction [60] and, consequently, also engagement [101] depend to a large extent on the sense of control over the environment and tasks performed. The sense of control is also reduced by stress at work [59] and protects against burnout. The same is true for the other predictors, i.e., management support and role definition. Many studies have indicated that management support reduces stress [62,63], promotes teamwork [64] and strengthens organisational culture [53], all of which are very important, especially in emergency situations, and the SARS-CoV-2 pandemic should be considered as such. Similarly, in the case of the Role variable, when it does not conflict with employees’ own value systems, it does not raise doubts in terms of how they should behave in a given situation when performing tasks. It is an important work resource. Therefore, it can be assumed that all three resources of the work situation (control, role, and management support) are also important in threatening situations.

In the case of remote work, the best-fit prediction model was obtained when all four groups of explanatory variables (a, b, c, and d) were taken into account. However, only some explanatory variables from groups b, c and d turned out to be predictors. As expected, it was assumed that if someone works longer in a given position, their engagement will not change during a pandemic that requires remote work. Most likely, even before the pandemic, these employees had experienced remote work and their engagement during the pandemic did not decrease but rather increased. Employees in this group had already developed a positive attitude towards this form of work and the pandemic only strengthened it. This model is consistent with another predictor, which concerns the attitude towards remote work (d), in particular the perception of the temporal dimension of remote work. Nowadays, it is natural to use advanced digital technologies at work. It can be assumed that this form of work is treated by employees as a privilege, as has been found in several studies [7,8,10,11,12,13].

Another important predictor was the fact that employees had social network accounts. People are social creatures and social isolation emphasises this need even more. This fact has been pointed out in several studies [102]. Nowadays, one substitute for meeting this need for daily contact is virtual contact through social media. Several studies have shown that being active on social networks relieves stress and improves the quality of life. Continuously checking emails, especially after working hours, also turned out to be a negative predictor which significantly reduces employees’ engagement in the performance of their professional duties. This fact has already been pointed out in other studies. This factor seems to be particularly important in the situation of remote work performed at home, where the employee should have a precisely defined time structure divided into professional duties and rest. Otherwise, he has a feeling of discomfort and lack of privacy and rest. This has also been emphasised by other researchers, who indicated that for this reason employees negatively assessed work performed remotely [10,11,28]. When it comes to work resources, only two variables turned out to be predictors of engagement. A positive work engagement predictor was the Control variable that was assumed in this study. On the other hand, the Relationships variable turned out to be a strong negative predictor of work engagement. When it comes to Control, the correlation is logical. The Relationships variable has an exceptionally strong impact on well-being and, as a result, on work engagement, especially when the relationships are negative. Reduced engagement in remote work as a result of poor relationships with colleagues and management can be both an effect and a cause. Several studies have indicated that, during the SARS CoV-2 pandemic, employees experienced job insecurity [103], pay inequality [14], a feeling of unequal burden of duties [23], more exposure to the threat of coronavirus infection and harassment [21], and even deportation in the case of immigrants [21,22]. These factors can lead to conflicts at work.

In the case of work performed in hybrid mode (alternately on-site and remote), the highest percentage of explanatory variations was obtained after excluding the variables from the analysis that describe the attitude towards remote work (d). Then, two explanatory variables, Control and Role, i.e., the variables from group (c), turned out to be work engagement predictors. These results are logical. Having control of the work performed by the employee is important for work engagement, regardless of whether work is performed on-site or remotely. Similarly, in the case of the Role variable, the employee knows what is expected of him and how tasks should be performed. Both these variables are most conducive to work engagement when working in a hybrid manner. Thus, it seems that women show greater engagement in the hybrid form of work, although these results are only on the edge of the statistical trend. The lack of predictive value of the variables from block (d) in the case of hybrid work may be an indication that these results should be analysed in both a positive and negative aspect. In the first case, remote work is perceived as a privilege and applies particularly to employees who also worked remotely before the pandemic [7,8,10,11,12,13]. In the second case (negative assessment), employees may have experienced a sense of injustice: why are they the only ones working on an on-site basis while others are working in a hybrid form? Why do managers contact some employees face to face and others only by email? The only factors that do not raise such doubts in our study are control and role.

## 5. Conclusions

The conducted study has not shown any differences in terms of any type of work engagement, regardless on the form in which it is performed. As for the characteristics of work, it is related to only two variables: Control and Relationships. In the case of Control, the strongest relations occur with respect to the hybrid and remote modes. In the case of the Relationships variable, the strongest relations occur with respect to the hybrid and on-site modes. In turn, the attitude towards remote work is related to each of the forms of work implementation in the situation of the SARS-CoV-2 epidemic. Employees who work remotely on a daily basis perceive the most positive, negative, and temporal aspects of remote work. The temporary aspect of remote work is also noticed by employees who perform their professional duties in hybrid form.

When looking for the predictors of work engagement in the SARS-CoV-2 situation, it was found that they differ depending on the current form of work. In the case of on-site work, its predictors are only factors related to the work situation (Control, Role, and Management Support), similarly as in the case of work performed in a hybrid manner (Control and Role). A broader list of predictors was obtained only in the case of performing work remotely, which is most beneficial from the point of view of protection against COVID-19 infection. These include not only the variables related to the characteristics of the work environment (control, relations), but also demographic variables (duration of remote work), social conditions (presence on social media and the employer’s respect for working hours), and attitude towards remote work (in particular, the belief that it is something natural in an emergency).

## 6. Limitations

The limitation of the study is the fact that it was carried out online. Many respondents could therefore falsify their data or present false opinions. Since the study was carried out at the peak of COVID-19 infections (the third wave in Poland), we may assume that the state of social isolation also translated into the evaluation of every form of work, in particular remote work. Another consequence of the pandemic situation was the reduced control over the composition of the group. This resulted in a certain inadequacy of the group composition in terms of the demographic characteristics controlled in the study. However, in a situation of a strict sanitary regime and online research, this problem could not be solved otherwise.

Another limitation is related to the size of the group. Although the total size is acceptable, the group sizes are not large when divided into remote, hybrid, and on-site workers. In future research, it should be useful to further increase the number of surveyed participants. A significant limitation of the research was also the deliberate selection of the subjects.

## Figures and Tables

**Table 1 ijerph-19-02400-t001:** Frequencies of describing the working conditions of remote work (*n* = 533).

Do My Housing Resources Allow Me to Comfortably Work at Home?	*n*	%
Definitely not	33	6.2
Probably not	54	10.1
Yes and no	123	23.1
Probably yes	181	34.0
Definitely yes	142	26.6
Internet connection I have at home:	*n*	%
Low bandwidth	57	10.7
Low bandwidth shared with others	84	15.8
Good bandwidth	376	70.5
I have no internet	16	3.0

**Table 2 ijerph-19-02400-t002:** Means and deviations as well as the analysis of variance of work engagement in individual groups of respondents distinguished according to the form of work.

Group of Workers		UWES Dedication		UWES Vigor,		UWES Absorption		UWES Total Work Engagement	
	*n*	M	SD	M	SD	M	SD	M	SD
Remote	139	10.26	3.30	11.55	3.21	10.87	3.10	32.68	8.33
Hybrid	140	10.24	3.04	11.52	3.22	11.33	2.85	33.09	8.03
On-site	254	10.66	3.57	11.25	3.64	10.70	3.46	32.61	9.03
Total	533	10.44	3.37	11.40	3.42	10.91	3.22	32.75	8.58
F		0.478		1.00		1.733		0.145	
Df		2/530		2/530		2/530		2/530	
*p*		n.i.		n.i.		n.i.		n.i.	

n.i.—not important.

**Table 3 ijerph-19-02400-t003:** Means and deviations as well as analysis of the variance of work characteristics in individual groups of respondents distinguished according to the form of work.

Variable:	Work Mode:	Remote	Hybrid	On-Site	F	DF	*p*
Demands	M	21.96	22.06	21.81	0.088	2/530	n.i.
	SD	6.08	5.2	5.96			
Control	M	22.09	22.7	21.26	4.621	2/530	0.01
	SD	3.9	4.4	5.09			
Management Support	M	16.27	16.19	15.7	2.187	2/530	n.i.
	SD	2.94	2.46	3.16			
Colleagues’ Support	M	13.56	13.83	13.5	1.144	2/530	n.i.
	SD	2.28	1.68	2.24			
Relationships	M	8.24	7.72	8.54	3.106	2/530	0.046
	SD	3.43	2.6	3.17			
Role	M	19.73	20.55	20.13	2.014	2/530	n.i.
	SD	3.05	3.05	3.74			
Change	M	10.61	10.58	10.51	0.095	2/530	n.i.
	SD	2.14	1.92	2.53			
	*n*	139	140	254			

n.i.—not important.

**Table 4 ijerph-19-02400-t004:** Means and deviations as well as the analysis of variance of work engagement in individual groups of respondents distinguished according to the form of work.

Group of Workers		TPZ Positive Reinforcement		TPZ Negative Reinforcement		TPZ Temporal Orientation	
	*n*	M	SD	M	SD	M	SD
Remote	139	12.61	2.96	9.35	2.15	12.62	2.39
Hybrid	140	11.23	3.26	8.39	1.83	12.34	1.79
On-site	254	11.59	2.96	7.78	2.29	11.08	2.51
Total	533	11.76	3.08	8.35	2.23	11.81	2.41
F		7.99		24.13		24.83	
Df		2/530		2/530		2/530	
*p*		0.001		0.001		0.001	

**Table 5 ijerph-19-02400-t005:** Regression coefficients for explanation of work engagement for the group of remote workers.

Variables	Model 1			Model 2			Model 3			Model 4		
	β	*p*	r_a(b,c)_	β	*p*	r_a(b,c)_	β	*p*	r_a(b,c)_	β	*p*	r_a(b,c)_
Sex	0.017	0.837	0.017	0.024	0.771	0.024	−0.003	0.972	−0.003	0.020	0.790	0.019
Age	−0.202	0.113	−0.130	−0.181	0.161	−0.115	−0.118	0.331	−0.069	−0.091	0.451	−0.053
*position*	0.180	0.035	0.174	0.175	0.045	0.165	0.105	0.198	−0.092	0.157	0.061	0.133
How long have you been working in your current position?	0.372	0.004	0.243	0.335	0.011	0.211	0.269*	0.023	0.164	0.256	0.029	0.155
Do your supervisors contact you after working hours?				0.074	0.424	0.066	0.003	0.969	0.003	−0.022	0.796	−0.018
Do you read work emails after working hours?				−0.135	0.151	−0.118	−0.179	0.032	−0.154	−0.193	0.020	−0.165
Do you have any social media accounts?				0.046	0.583	0.045	0.249	0.006	0.200	0.222	0.014	0.175
HSE Demands							−0.027	0.805	−0.018	−0.016	0.887	−0.010
HSE Control							0.253	0.006	0.200	0.191	0.040	0.146
HSE Management support							0.044	0.708	0.027	0.057	0.628	0.034
HSE Colleagues Support							0.171	0.127	0.109	0.157	0.158	0.100
HSE Relationships							−0.323	0.001	−0.236	−0.347	0.001	−0.246
HSE Role							0.142	0.121	0.111	0.162	0.075	0.126
HSE Change							−0.092	0.406	−0.059	−0.135	0.225	−0.086
TPZ Positive reinforcement										0.127	0.102	0.116
TPZ Negative reinforcement										−0.017	0.837	−0.015
TPZ Temporal orientation										0.161	0.056	0.135
AdjR^2^	0.080			0.077			0.301			0.320		
Change R^2^	0.107			0.018			0.247			0.032		
F of change	3.994			0.874			6.973			2.163		
df1	4.000			3.000			7.000			3.000		
df2	134.00			131.00			124.00			121.00		
*p* of change	0.004			0.456			0.000			0.096		

**Table 6 ijerph-19-02400-t006:** Regression coefficients for explanation of work engagement for the group of hybrid employees.

Variables	Model 1			Model 2			Model 3			Model 4		
	β	*p*	r_a(b,c)_	β	*p*	r_a(b,c)_	β	*p*	r_a(b,c)_	β	*p*	r_a(b,c)_
Sex	−0.200	0.020	−0.196	−0.197	0.020	−0.193	−0.149	0.076	−0.140	−0.142	0.098	−0.131
Age	0.174	0.159	0.118	0.232	0.065	0.153	0.091	0.490	0.054	0.088	0.521	0.051
position	0.156	0.076	0.148	0.178	0.041	0.169	0.097	0.285	0.084	0.110	0.239	0.093
How long have you been working in your current position?	−0.157	0.194	−0.108	−0.160	0.181	−0.110	−0.069	0.568	−0.045	−0.071	0.562	−0.046
Do your supervisors contact you after working hours?				0.092	0.328	0.080	−0.047	0.628	−0.038	−0.038	0.704	−0.030
Do you read work emails after working hours?				−0.186	0.051	−0.161	−0.111	0.240	−0.092	−0.111	0.245	−0.092
Do you have an account on social networks?				−0.149	0.095	−0.138	−0.140	0.107	−0.127	−0.135	0.124	−0.122
HSE Demands							0.042	0.670	0.033	0.046	0.662	0.035
HSE Control							0.206	0.049	0.155	0.205	0.056	0.152
HSE Management support							0.115	0.295	0.082	0.116	0.299	0.082
HSE Colleagues Support							−0.121	0.258	−0.089	−0.114	0.289	−0.084
HSE Relationships							−0.072	0.498	−0.053	−0.078	0.475	−0.056
HSE Role							0.213	0.037	0.164	0.217	0.036	0.167
HSE Change							0.031	0.769	0.023	0.029	0.785	0.022
TPZ Positive reinforcement										−0.029	0.762	−0.024
TPZ Negative reinforcement										0.010	0.910	0.009
TPZ Temporal orientation										0.091	0.311	0.080
AdjR^2^	0.042			0.068			0.152			0.139		
Change R^2^	0.070			0.045			0.122			0.007		
F of change	2.522			2.261			2.855			0.391		
df1	4			3			7			3		
df2	135			132			125			122		
*p* of change	0.044			0.084			0.009			0.759		

**Table 7 ijerph-19-02400-t007:** Regression coefficients for explanation of work engagement for the group of on-site workers.

Variables	β	*p*	r_a(b,c)_	β	*p*	r_a(b,c)_	β	*p*	r_a(b,c)_	β	*p*	r_a(b,c)_
Sex	−0.121	0.053	−0.120	−0.117	0.061	−0.116	−0.094	0.077	−0.091	−0.095	0.074	−0.092
Age	−0.055	0.477	−0.044	−0.052	0.518	−0.040	−0.005	0.944	−0.004	0.000	0.998	0.000
position	0.174	0.007	0.168	0.165	0.011	0.157	0.069	0.215	0.064	0.066	0.248	0.059
How long have you been working in your current position?	0.124	0.102	0.101	0.098	0.200	0.079	0.073	0.257	0.058	0.069	0.293	0.054
Do your supervisors contact you after working hours?				−0.107	0.105	−0.100	−0.080	0.157	−0.073	−0.076	0.187	−0.068
Do you read work emails after working hours?				−0.073	0.279	−0.067	−0.043	0.456	−0.038	−0.043	0.460	−0.038
Do you have any social media accounts?				0.063	0.366	0.056	0.030	0.611	0.026	0.021	0.718	0.019
HSE Demands							−0.013	0.870	−0.008	−0.014	0.868	−0.009
HSE Control							0.331	0.000	0.269	0.334	0.000	0.271
HSE Management support							0.180	0.019	0.121	0.170	0.028	0.113
HSE Colleagues Support							−0.005	0.943	−0.004	−0.002	0.973	−0.002
HSE Relationships							−0.042	0.601	−0.027	−0.042	0.615	−0.026
HSE Role							0.268	0.000	0.203	0.275	0.000	0.206
HSE Change							−0.054	0.505	−0.034	−0.050	0.545	−0.031
TPZ Positive reinforcement										−0.016	0.821	−0.012
TPZ Negative reinforcement										0.079	0.223	0.063
TZP Temporal orientation										−0.067	0.347	−0.048
AdjR^2^	0.037			0.048			0.335			0.333		
Change R^2^	0.052			0.022			0.298			0.005		
F of change	3.430			1.924			16.222			0.682		
df1	4			3			7			3		
df2	249			246			239			236		
*p* of change	0.009			0.126			0.000			0.564		

## Data Availability

Data available in the Institute of Applied Psychology Jagiellonian University, Department of Work and Organizational Psychology.

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
