# Peer review of "Remote, Hybrid, and On-Site Work during the SARS-CoV-2 Pandemic and the Consequences for Stress and Work Engagement"

_ijerph, 2022, doi:10.3390/ijerph19042400_

Round 1
Reviewer 1 Report
I was unsure of what method and design this study followed. It is possible to indicate from the start, as a benefit for the reading audience.
Could do with some condensing and conciseness. There is an abundance of information, the paper could almost be two manuscripts.
References need to be updated and include more current research studies.
Many minor grammatical and punctuation errors. Suggest running the document through a grammar check platform to catch some of them.
Line 27 refers to ILO as a reference. Is there an indication of what the letters mean?
Line 73 references are listed with spaces and semi-colons. Line 92 shows no space. Line 96 shows a space. Maintaining consistency in the body of the manuscript, based on style (APA, MLA, CMOS, AMA, etc.) is a suggestion. Line 100 list with no separation.
Line 111: is ICT mentioned earlier?
Line 129-130: spell out numbers 10 and under: five years; two and a half times…what do the numbers (28.937: 11, 708) represent?
For ease of visibility, listing all the hypotheses in one place would be beneficial, so the readers will know what is being tested, how that flows together, what the results show, and recommendations for further studies if any.
Line 195: we hypothesized that the stress of remote workers would be worse than that of on-site workers (H32) what number is this for?
Line 406: please show those H numbers. In order to verify these hypotheses (Hx, Hx, etc.).
Author Response
At the outset, we would like to thank everyone involved in the review process for their efforts and the opportunity to refine the publication. Thanks to the comments received, we have improved the text in accordance with the suggestions indicated by the reviewers. Each of reviewer is commented separately.
|
Rev 1 |
Our reply |
|
I was unsure of what method and design this study followed. It is possible to indicate from the start, as a benefit for the reading audience. |
Thank you for bringing this aspect to our attention. We have added information about the project in the abstract that “cross-sectional questionnaire study was conducted”. Thanks to this the reader will be informed what is the method and design of this study |
|
Could do with some condensing and conciseness. There is an abundance of information, the paper could almost be two manuscripts. |
Thank you for this valuable comment. The text was reviewed once again and shortcuts were made eliminating side threads. In addition, the structure of the theoretical introduction was reorganized. Now, each variable has its own section. We hope that this will make the text easier to receive for the reader. |
|
References need to be updated and include more current research studies. |
Thank you for this comment. The literature is 56% from the last 5 years (2016+), the rest are links to classic publications. The literature has been updated - a few links to publications from the last two years have been added, although the classic entries remain. Since the literature is updated automatically (Mendeley) this was not indicated in the bibliography. |
|
Many minor grammatical and punctuation errors. Suggest running the document through a grammar check platform to catch some of them. |
Thank you for bringing this aspect to our attention. A proof editor (native speaker) has re-reviewed the text and removed grammatical and punctuation errors. |
|
Line 27 refers to ILO as a reference. Is there an indication of what the letters mean? |
Thank you for noticing this detail. We have corrected the defect by writing the full name: International Labour Organization. |
|
Line 73 references are listed with spaces and semi-colons. Line 92 shows no space. Line 96 shows a space. Maintaining consistency in the body of the manuscript, based on style (APA, MLA, CMOS, AMA, etc.) is a suggestion. Line 100 list with no separation. |
Thank you for your attention to help us improve the quality of the text. The appearance of references in the text has been standardized. |
|
Line 111: is ICT mentioned earlier? |
Thank you for this comment. The abbreviation ICT is commonly used to denote the Information and Communication Technologies. The full name has been added in the text for clarity of understanding. |
|
Line 129-130: spell out numbers 10 and under: five years; two and a half times…what do the numbers (28.937: 11, 708) represent? |
Thank you for the opportunity to clarify the content of the article. The quoted numbers express the number of publications that appeared in each period. The dates of the period considered for counting the publications are added in brackets. |
|
For ease of visibility, listing all the hypotheses in one place would be beneficial, so the readers will know what is being tested, how that flows together, what the results show, and recommendations for further studies if any. |
We thank the reviewer for pointing this out. We agree with it, which is why the Hypotheses are collected in one place and preceded by an introduction: The presentation of the research method requires a clear formulation of the research focus and problem. The research question of our project is whether the groups (remote, hybrid and one-site work) differ in terms of work engagement, stress, and the links between these variables. For this purpose, all the hypotheses posed and justified in the theoretical introduction are collected at the beginning of this section. This makes it easier to understand what is being researched and to see that the research method is well suited to the problem. The subject of our empirical analysis is to check the validity of the following four main hypotheses and some of their details: And than we listed all hypothesis once more |
|
Line 195: we hypothesized that the stress of remote workers would be worse than that of on-site workers (H32) what number is this for? |
We apologise for the error we made and thank the reviewer for pointing it out. Hypothesis H3 - it has been corrected. |
|
Line 406: please show those H numbers. In order to verify these hypotheses (Hx, Hx, etc.). |
The interpretation of the truth of the hypotheses is described in the discussion section. Each hypothesis discussed has a reference to its number. Therefore, marking the numbers of hypotheses in the results section is considered unnecessary. Moreover, some statistics serve to verify several specific hypotheses and references to them will obscure the message. |
Reviewer 2 Report
The authors have addressed a very important topic in occupational health. Studying the work environment itself when the work is shifted to be remotely is very important, since there are differences in the work settings that will affect the worker.
The introduction provides a very good background of the topic, discussing the different perspectives of remote work and its consequences on the workers and the business itself. However, I would suggest that the authors shorten the introduction section and keep it more concise and focused on describing the model that was used in the methods.
It would also be good if the authors provide some of the most interesting results in the abstract with some more details in terms of numbers and percentages.
Author Response
At the outset, we would like to thank everyone involved in the review process for their efforts and the opportunity to refine the publication. Thanks to the comments received, we have improved the text in accordance with the suggestions indicated by the reviewers. Each of reviewer is commented separately.
Rev.2
|
Rev 2 |
Our reply |
|
The authors have addressed a very important topic in occupational health. Studying the work environment itself when the work is shifted to be remotely is very important, since there are differences in the work settings that will affect the worker. |
Thank you for your positive opinion |
|
The introduction provides a very good background of the topic, discussing the different perspectives of remote work and its consequences on the workers and the business itself. However, I would suggest that the authors shorten the introduction section and keep it more concise and focused on describing the model that was used in the methods. |
Thank you for your positive opinion |
|
It would also be good if the authors provide some of the most interesting results in the abstract with some more details in terms of numbers and percentages. |
Thank you for this comment and the opportunity to improve the quality of the text. More details about results of regression analysis in the form of information about which variables were statistically significant in the equation in each group were introduced in the abstract. Introducing the coefficient values I think could cloud the picture rather than clarify it and therefore were not added. |
Best regards,
Bohdan Rożnowski
Reviewer 3 Report
The paper is well organized and structured. The work engagement is in continuous transformation, and new processes are implemented oriented on remote work. In this direction I suggest to introduce in the introduction section the new technologies which could support the work management such as: chatbot, and big data and artificial intelligence for KPI worker evaluation.
The strength of the paper is the questionnaire based approach and results highlighting internet activity, and regression coefficient evaluation.
The paper weakness is that about my opinion "A total of 544 workers were surveyed" are few for a final decision about work engagement trend.
For future works I suggest to further increase the number of surveyed.
Author Response
Dear Sir/Lady,
At the outset, we would like to thank everyone involved in the review process for their efforts and the opportunity to refine the publication. Thanks to the comments received, we have improved the text in accordance with the suggestions indicated by the reviewers. Each of reviewer is commented separately.
Rev.3
|
Rev 3 |
Our reply |
|
The paper is well organized and structured. The work engagement is in continuous transformation, and new processes are implemented oriented on remote work. In this direction I suggest to introduce in the introduction section the new technologies which could support the work management such as: chatbot, and big data and artificial intelligence for KPI worker evaluation. |
Thank you for this interesting comment. I agree about the connections pointed out by the reviewer, but in our project the main focus was on the determinants of work engagement for remote workers. The idea proposed by the third reviewer to add content about chatbot and big data would involve expanding the text, which is in contrast to the first reviewer's suggestion to shorten the text. Moreover, the research method did not include content related to new technology issues and we would not be able to address the proposed expansion of the content in the empirical section |
|
The strength of the paper is the questionnaire based approach and results highlighting internet activity, and regression coefficient evaluation. |
Thank you for your positive feedback on our work |
|
The paper weakness is that about my opinion "A total of 544 workers were surveyed" are few for a final decision about work engagement trend. For future works I suggest to further increase the number of surveyed. |
Thank you for pointing out this aspect. Of course, increasing the sample size would improve the reliability of the study. However, in view of the dynamic change of the Pandemic situation it is not possible to enlarge it. We have added in the section on limitation the remark indicated by the reviewer. It would indeed be better if the research sample was larger. |
Best regards,
Bohdan Rożnowski
